# Effect of a Short-Time Probiotic Supplementation on the Abundance of the Main Constituents of the Gut Microbiota of Term Newborns Delivered by Cesarean Section—A Randomized, Prospective, Controlled Clinical Trial

**DOI:** 10.3390/nu12103128

**Published:** 2020-10-14

**Authors:** Joanna Hurkala, Ryszard Lauterbach, Renata Radziszewska, Magdalena Strus, Piotr Heczko

**Affiliations:** 1Neonatology Clinic, University Hospital, Medical College, Jagiellonian University, Kopernika 23 Str, 31-501 Kraków, Poland; joannahurkala@yahoo.com (J.H.); ryszard@lauterbach.pl (R.L.); rradziszewska@wp.pl (R.R.); 2Chair of Microbiology, Medical College, Jagiellonian University, Kraków, Czysta 18 Str, 31-121 Kraków, Poland; mbstrus@cyf-kr.edu.pl

**Keywords:** neonates, gut microbiota, cesarean section, probiotic, dysbiosis

## Abstract

The gut microbiota plays a pivotal role in the maintenance of human health. Numerous factors, including the mode of delivery, impact early gut colonization in newborns. Recent research focuses on the use of probiotics in the prevention of gut dysbiosis in newborns delivered by cesarean section (CS). The objective of this study was to determine whether a probiotic supplement given to newborns delivered by CS during their stay in the maternity ward alters the pattern of early gut colonization by lactic acid bacteria versus potential pathogens. A prospective, randomized trial was conducted. In total, 150 newborns, born at 38–40 weeks gestational age and delivered by CS, were included in the study. They were randomized into the intervention group, supplemented orally with a probiotic containing *Bifidobacterium breve* PB04 and *Lactobacillus rhamnosus* KL53A, and the control group. Stool samples were obtained on days 5 and 6 of life and after one month of life and were analyzed for the presence and abundance of the main groups of bacteria. An application of two probiotic bacteria during the first days of life after CS resulted in quick and abundant colonization by days 5 and 6, with high populations of *L. rhamnosus* and *B. breve*. The applied bacterial strains were present in the majority of neonates one month after. The supplementation of term neonates delivered by cesarean section immediately after birth with a mixture of *L. rhamnosus* and *B. breve* enriched the gut microbiota composition with lactic acid bacteria.

## 1. Introduction

The early development of the intestinal microbiota in neonates born vaginally starts at birth due to the acquisition of organisms from the vaginal microbiota, but also from other maternal sources (i.e., skin) and environmental sources [1]. The next step enabling the early establishment of the microbiota is breastfeeding [2]. Thus, the neonatal gut microbiota develops by the sequential colonization of intrauterine/vaginal birth-associated taxa, such as *Lactobacillus* and *Bifidobacterium*, skin-derived taxa, and other typical early colonizers from the environment, such as *Streptococcus* and *Enterococcus*, and potential pathogens (*Staphylococcus*, *Enterobacteriaceae*) usually detected in children born in hospitals. As shown in our recent review [3] and study on neonatal sepsis [4], both Gram-negative rods and staphylococci originating from gut microbiota are able to translocate from the gut to the bloodstream and cause sepsis.

The establishment of the gut microbiota may be affected and disturbed by cesarean section (CS) delivery, the method of feeding, antibiotic treatment, gestational age, and environmental factors. Among them, CS is suggested to be one of the major factors in the early-life disruption of neonatal gut microbiota [5]. In particular, the acquisition of *Lactobacillus* is significantly impaired in CS-born infants compared with those delivered vaginally [6]. Moreover, at the colonization level, the genera *Bifidobacterium* and *Bacteroides* are significantly more prevalent in vaginally delivered infants compared with those delivered by CS [7].

There are observations suggesting that early-life aberrations in gut microbiota may have long-lasting consequences that have been associated with an increased risk of developing metabolic or immunological diseases in later life [1,8,9,10]. An immediate risk of infection (including necrotizing enterocolitis and sepsis) resulting from an altered gut microbiota (dysbiosis) has also been reported in very low birth weight (VLBW) neonates [11,12].

Therefore, attempts to modulate the gut microbiota in neonates delivered by CS with probiotics representing bacterial species from the vagina have been made, with the aim of achieving a bacterial composition more similar to that present in vaginally delivered children, i.e., by increasing proportions of *Lactobacillus* and *Bifidobacterium* versus potential pathogens in the gut microbiota. Although different probiotic preparations have been used, a higher abundance of *Lactobacillus* or *Bifidobacterium* was consistently detected in fecal samples [13,14].

## 2. Aim

Since the gut microbiota of the neonates delivered by CS is characterized by low numbers of lactic acid bacteria and high numbers of the potential pathogens, the aim of this study was to assess the effects of probiotic supplementation on the composition of the main constituents of the gut microbiota; that is, populations of lactobacilli and bifidobacteria in comparison to those of the potential pathogens in healthy children delivered by CS. This was studied by comparing the colonization rates based on testing fecal samples taken on the day of discharge from hospital and after one month of life. The additional aim was to compare colonization rates in this group of neonates with a group of neonates with low birth rates previously tested by us using the same probiotic preparation, in order to follow the gut colonization by probiotic bacteria in neonates with immature and mature gut epithelium.

## 3. Methods

A prospective, randomized trial, including a control group, was conducted in the Neonatal Clinic of Jagiellonian University Hospital in Kraków from April 2014 to April 2017. In total, 150 newborns, born at 38–40 weeks gestational age by CS, were included in the study. In all cases, CS was performed due to implications (e.g., previous cesarean delivery, breech presentation, dystocia, or multiple gestation) rather than an emergency. The investigations were carried out following the rules of the Declaration of Helsinki of 1975 (https://www.wma.net/what-we-do/medical-ethics/declaration-of-helsinki/), revised in 2013.

Written consent was obtained from the parents of all children prior to recruiting their newborns. 

The recruited neonates were randomized into two groups: the intervention group and the control group. Newborns in the intervention group were supplemented with a probiotic product containing lyophilized *Bifidobacterium breve* PB04 and *Lactobacillus rhamnosus* KL53A (FFbaby^®^, in the form of openable capsules, kindly donated by the producer: IBSS BIOMED S.A., Krakow, Poland). It was administered orally within the first hour of life and then every 12 h, the contents of which were dissolved in the mother’s milk or formula after opening the capsule. This was done up until discharge on day 5 or 6 after delivery. The total number of probiotic bacteria was 2 × 10^6^ colony forming units (CFU) per day. This trial received ethical approval from the Independent Ethics Committee of Jagiellonian University, Krakow, Poland (no. KBET/46/B/2014 on 27 March 2014). The trial was also registered in ClinicalTrials.gov no. NCT03657485. 

The strains (*L. rhamnosus* KL53A and *B. breve* PB04) were well characterized in in vitro and in vivo studies (patent No. P406051). The strains included in the product have a documented human origin, having been isolated from the feces of healthy, breastfed neonates. Their strong anti-inflammatory, antipathogenic activity, high adherence ability to human Caco-2 and HT-29MTX cell lines, ability to survive in low gastric pH and bile, and tight-junctions stimulating properties, were confirmed in laboratory studies. The safety, probiotic properties, and gut mucosa colonizing ability of the strains were checked on gnotobiotic (germ-free) mice and rat neonates [15,16].

The colonization of the gut of preterm neonates and the safety of FFbaby^®^ was also confirmed in a randomized, placebo-controlled, multi-center clinical trial [17]. It found a correlation between the presence of *Bifidobacterium breve* and a lower frequency of staphylococcal sepsis episodes. No episodes of sepsis caused by the probiotic bacteria contained in the product were noted.

The inclusion criteria for the study were as follows: 38–40 weeks gestational age, normal, healthy pregnancy, delivery by CS, good clinical condition after birth, Apgar scale: 8–10 points, proper gestational mass >2500 g, and the informed consent of the parents. The exclusion criteria were as follows: the conditions mentioned above not being fulfilled, a lack of informed consent by the parents, or their resignation from the study.

Mothers were encouraged to breastfeed their babies during hospitalization and thereafter. 

Stool samples were initially obtained on days 5 or 6 of life and then again after one month of life. Parents were instructed to collect a stool sample from the diaper and transfer the sample immediately into transport media (three 15 mL bottles containing liquid media: Schaedler broth (Difco Laboratories, Detroit, MI, USA) with 15% glycerol, MRS (Oxoid, Basingstoke, UK) with 15% glycerol, TPY (Difco) with 15% glycerol), then samples were stored at −70 °C prior to analysis. 

Before microbiological testing, the samples were defrosted at room temperature, weighed, and minced by a stomacher (BagMixer 400 P, Interscience, Saint Nom, France). Serial dilutions in Schaedler broth (Difco) were plated onto different agar media to isolate bacterial genera/species representing the early gut microbiota, i.e., *Lactobacillus* and *Bifidobacterium*, as well as potentially pathogenic organisms such as *Clostridium*, *Staphylococcus*, *Enterococcus*, *Klebsiella*, and *Escherichia coli*. The following agar media were used: McConkey agar (Oxoid) for members of the Enterobacteriaceae, Columbia blood agar (Difco) with 5% sheep blood for staphylococci and streptococci, Enterococcosel agar (BBL/BD, Franklin Lakes, NY, USA) for enterococci, Sabourand agar (Oxoid) for *Saccharomyces*, MRS (De man, Rogosa and Sharpe) agar (Oxoid) for lactic acid bacteria, TPY agar (Oxoid) for *Bifidobacteria*, and Wilkins–Chalgren agar (Oxoid) for *Clostridia*. Then, the plates were incubated at 37 °C under aerobic and/or anaerobic conditions for 24 or 48 h. Bacterial colonies were counted and then representative colonies were isolated and identified using API sets (BioMerieux, Marcy l’Etoile, France): API 20E, API 20A, API Strep, API Staph, and API 50CHL. The final results were expressed as colony forming units/g of stool (CFU/g).

To assess the colonization of the infants’ gut with lactobacilli and bifidobacterial of the same species as administered during the study, representative colonies of *Lactobacillus* and *Bifidobacteria* grown on specific media were selected, subcultured, and identified using genotyping methods (PCR). The species-specific PCR for *L. rhamnosus* was performed using the following primers: PrI 5ʹ CAG ACT GAA AGT CTG ACG G 3ʹ and RhaII 5ʹ GCG ATG CGA ATT TCT ATT ATT 3ʹ, which amplifies a 190-bp fragment. The PCR conditions were as follows: one cycle at 92 °C for 2 min; then 30 cycles at 95 °C for 30 s, 55 °C for 30 s, and 72 °C for 30 s; followed by one cycle at 72 °C for 1 min. The following primers IDB31F 5ʹ TAG GGA GCA AGG CAC TTT GTG T 3ʹ and IDBC1R 5ʹ ATC CGA ACT GAG ACC GGT T 3ʹ were used to identify *B. breve*. The size of the PCR product was 827 bp and the protocol was as follows: one cycle at 94 °C for 5 min; then 35 cycles at 94 °C for 30 s, 64 °C for 40 s, and 72 °C for 30 s; followed by one cycle at 72 °C for 5 min [18,19].

The treatment allocation was conducted through a computer-generated randomization list. The results were statistically analyzed with Statistica software, version 11.0. Chi-quadrat Pearson’s test and the Mann–Whitney U test were used to analyze nonparametric data. For statistical analysis, numbers of bacterial cells belonging to different genera/species were grouped. 

## 4. Results

The total number of newborns included in the study was 148, as in two cases parents removed their infants from the study after initial recruitment. The intervention group comprised 71 newborns, and the control group comprised 77 newborns. Thirteen newborns were excluded from the intervention group after one month because the second stool sample was not provided; thus, the sample size for this group was 58. For the same reason, the final number of participants in the control group was 48, as shown in the participant flow chart (Table 1 and Figure 1).

Comparison between the groups showed that there were no significant differences with respect to sex, gestational mass, or maternal antibiotic treatment before labor.

Probiotic supplementation caused a highly significant increase in *Lactobacillus* numbers in the fecal samples of the intervention group compared with the control group, which harbored very low populations of lactobacilli (below 2 logs/g = 1 × 10^2^ c.f.u./g) (Figure 2).

The numbers of lactobacilli were high and varied from 6.8 to 8.5 logs per gram of feces. Molecular identification confirmed that the isolated lactobacilli belonged to the species *L. rhamnosus*. The numbers of *Bifidobacteria* in the feces of the probiotic intervention group were also high, ranging from 4.5 to 8.8 logs/g, which were significantly higher than those detected in the control infants (Figure 3). Molecular identification confirmed that the isolated *Bifidobacteria* belonged to the *B. breve* species. Low populations (cumulative counts between 2 and 6 logs/g) of potentially pathogenic cultivable bacteria (*Clostridium*, *Staphylococcus*, *Enterococcus*, *Klebsiella*, and *Escherichia coli*) were found in the feces samples of both groups, with no significant differences detectable between the groups.

High numbers of *Lactobacillus* (median value of 10 logs/g) remained in the majority of the fecal samples of probiotic-supplemented infants one month after administration. The presence of *Lactobacillus* was also noted in control infants, with an abundance varying from 4.6 to 7.8 logs/g, but these levels were significantly lower than in the intervention group (Figure 4). The abundance of bifidobacteria in the feces of the supplemented neonates was low in some cases (median value of below 8 logs/g), thereby resulting in a similar median value to that of the control infants (6.7 logs/g), with no significant difference between these groups (Figure 5).

When comparing the abundance of potentially pathogenic bacteria, their numbers varied widely (0 to over 7 logs/g) among individual neonates in the few days after delivery but showed a high median value, whereas they were more uniform one month later (reaching 8 logs/g), and this difference was significant. The potentially pathogenic bacteria were more abundant in one-month-old babies, showing median values of about 2 logs/g higher than the earlier time point in both groups (Figure 6).

## 5. Discussion

This study was undertaken to monitor populations of lactic acid bacteria versus potential pathogens in the feces of neonates delivered by cesarean section using standard quantitative culture methods. Such an approach was considered superior to testing the whole gut microbiota or its specific phylogenetic groups using new generation sequencing, since the intention was to study the effects of probiotic supplementation. Moreover, low populations of the bacteria found in the first days after delivery by C-section seemed to be better followed using the classical way. Vaginally delivered neonates, naturally colonized with relatively high numbers of the lactic acid bacteria during labor, were not included in the study for the same reasons.

Our study confirmed that neonates born in hospitals by CS are practically devoid of *Lactobacillus* and *Bifidobacteria* in their gut microbiota up to days 5 and 6 after labor, since these bacteria are virtually undetectable (below 2 log/g) in control babies. By contrast, bacteria considered to be potential pathogens were present in both the control and intervention groups. This observation confirmed previous findings that neonates delivered by CS in hospitals are rapidly colonized by bacteria derived from hospital environments, although it cannot be excluded that a proportion of the bacteria (especially the coagulase-negative staphylococci) are transferred from their mother’s skin [1]. One month after birth, colonization with potential pathogens was more pronounced, which may reflect the natural process of acquisition of bacteria from the environment, since Gram-negative rods, coagulase-negative staphylococci, and enterococci formed the majority of this population. It should be mentioned here that the risk of infection from these bacteria is much lower for one-month-old infants than for newborns [20]. The gut colonization of these babies also involved the bacteria *L. rhamnosus* and *B. breve* from the same sources, which probably persisted in the gut microbiota of the probiotic-supplemented infants. However, the population of *B. breve* declined during this period, which was in accordance with other reports [21], showing that the abundance of the bifidobacteria population is usually lower when a more complex gut microbiota develops. The presence of the probiotic bacteria in the control babies may be explained by contamination from other probiotic-supplemented infants during their hospital stay, or it may result from the intake of probiotic bacteria at home.

The mother’s skin, especially the areolar region involved in breastfeeding, is an important source for microbiota acquisition by newborns and is very important in assembling neonatal gut microbiota [1]. This reservoir is obviously highly variable from mother to mother and depends on many different factors. Consequently, acquisition of bacteria and subsequent gut colonization progresses differently in individual children, which causes large variations in the bacterial numbers present in feces, as was observed in our study.

The presence of high numbers of *L. rhamnosus* and *B. breve* in the feces of infants within a few days of their administration indicates their successful colonization and confirms the findings of previous reports [13,14]. In our previous studies, we administered the same probiotic strains to very low and low birth weight neonates (VLBW and LBW) with the aim of protecting them from sepsis. Indeed, *B. breve* colonization correlated with a lower incidence of staphylococcal sepsis, irrespective of probiotic supplementation [17]. However, differences existed between the current study and our previous study. In our previous study, colonization took longer in the premature neonates and *Lactobacillus* counts increased in the samples collected from weeks 2 to 7, while a significant increase in *Bifidobacterium* counts was detected in weeks 2 and 3. These differences may be explained by incomplete maturation of the gut mucosa [18,19] or by previous antibiotic treatment of the VLBW or LBW neonates in intensive care units. In this study, colonization was observed a few days after starting supplementation and persisted in a proportion of the tested infants after one month.

It should be stressed here that gut colonization with lactobacilli and bifidobacterial, as proved by quantitative cultures, was achieved in our study in a few days and lasted for one month. Garcia Rodenas et al. [13] fed children in their study for 6 months with an *L.reuteri* strain and still, the abundance of *L.reuteri* was not significantly increased and bifidobacteria were not even encountered. Two explanations for these discrepancies can be offered: 1) the *L.reuteri* strain chosen for Garcia Rodenas et al.’s study was not capable of colonizing gut mucosa in neonates in comparison to our strain of *L.rhamnosus,* or 2) feeding with *L.reuteri* started too late after birth (72 h) when receptors for lactobacilli were already fixed by previous colonizers. It is known from animal studies that the window for early colonization is rather short [17]. Bazanella et al. [14] studied a mixture of three Bifidobacterium strains from birth to 1 year and also found that bifidobacterial supplementation did not alter microbial profiles, but all three species of bifidobacteria were detected in fecal samples at month 4 after birth, although at an unknown abundance. Their data are more similar to ours, since colonization with bifidobacteria was achieved and retained for months. Moreover, colonization with bifidobacteria in their studies was related to a lower abundance of *Bacteroides fragilis,* which is also a potential pathogen.

This, as well as our previous study [17], indicates that in order to achieve colonization with probiotic bacteria, it is necessary to start supplementation immediately after birth when the gut microbiota of a newborn is very scanty, especially in those born with C-section and with mucosa ready to acquire bacteria from the environment [20,21].

## 6. Conclusions

Supplementation of the neonates born by C-section with a mixture of *L. rhamnosus* and *B. breve* immediately after birth increases numbers of lactobacilli and bifidobacteria in their gut and thus it mimics the natural colonization of newborns born naturally.

## Figures and Tables

**Figure 1 nutrients-12-03128-f001:**
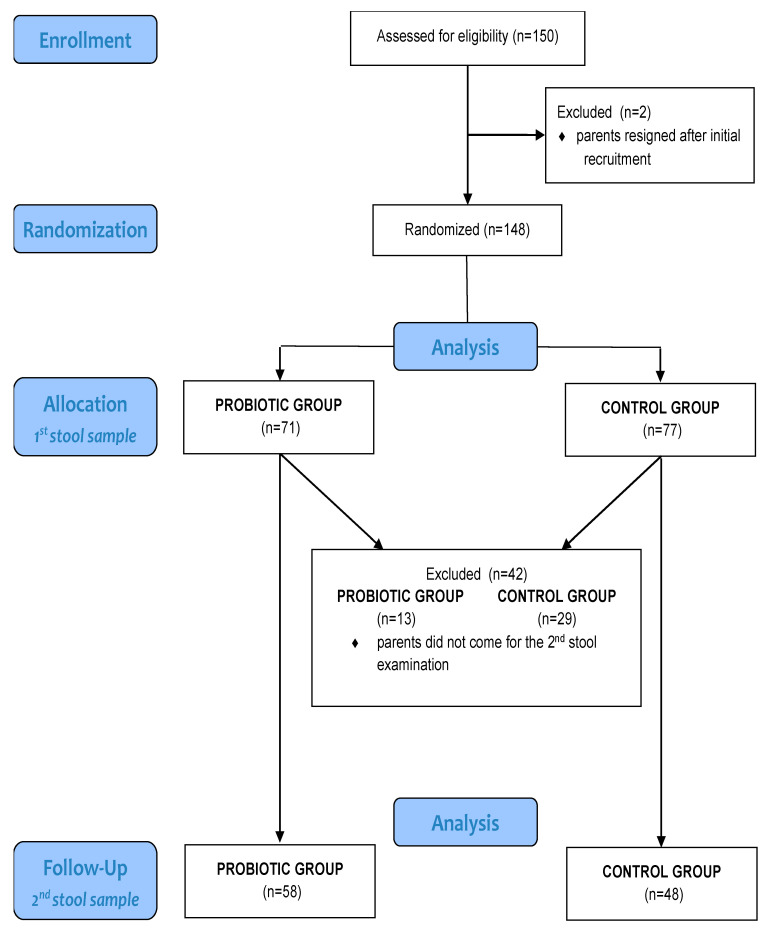
Participant flow chart.

**Figure 2 nutrients-12-03128-f002:**
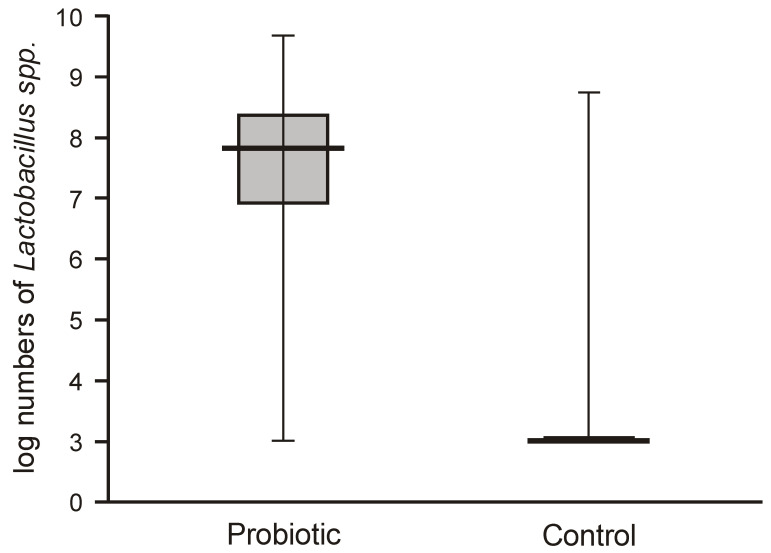
Abundance of lactobacilli in feces obtained from the first sampling of neonates supplemented with probiotics vs. controls (Mann–Whitney test Z = 8.9629; *p* < 0.00001)

**Figure 3 nutrients-12-03128-f003:**
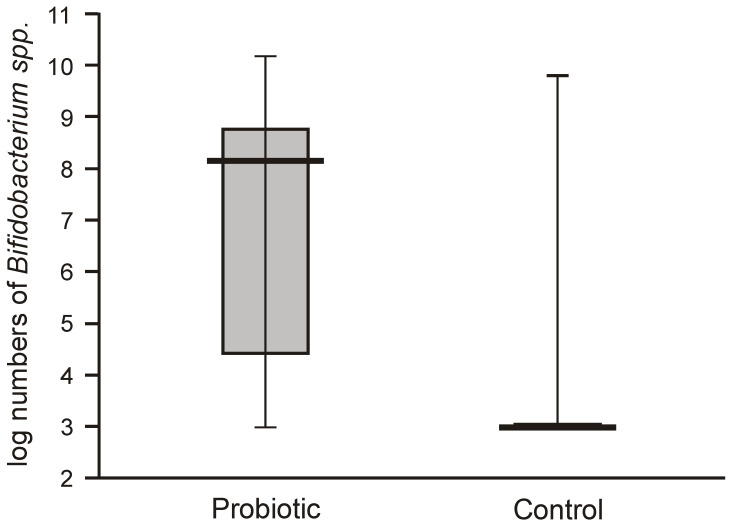
Populations of bifidobacteria in feces obtained from the first sampling of neonates supplemented with probiotics vs. controls (Mann–Whitney test Z = 7.7117; *p* < 0.00001).

**Figure 4 nutrients-12-03128-f004:**
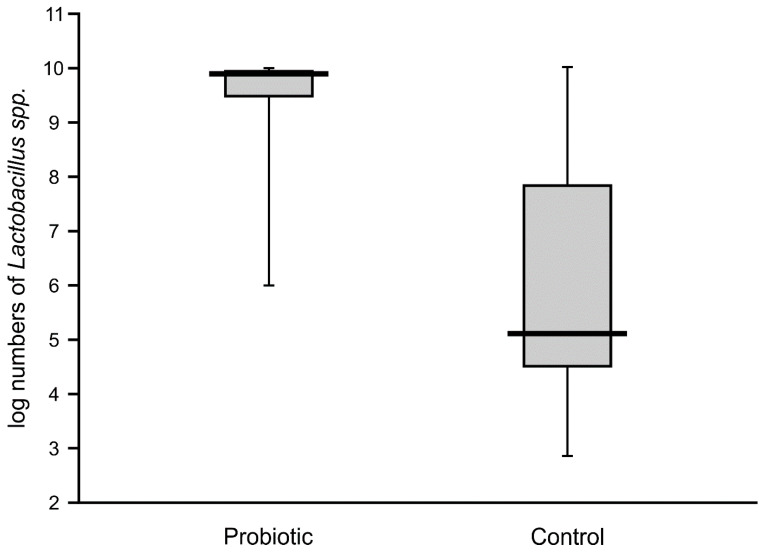
Abundance of lactobacilli in feces obtained from the second sampling of neonates supplemented with probiotics vs. controls (Mann–Whitney test, Z = 8.48069; *p* < 0.00001).

**Figure 5 nutrients-12-03128-f005:**
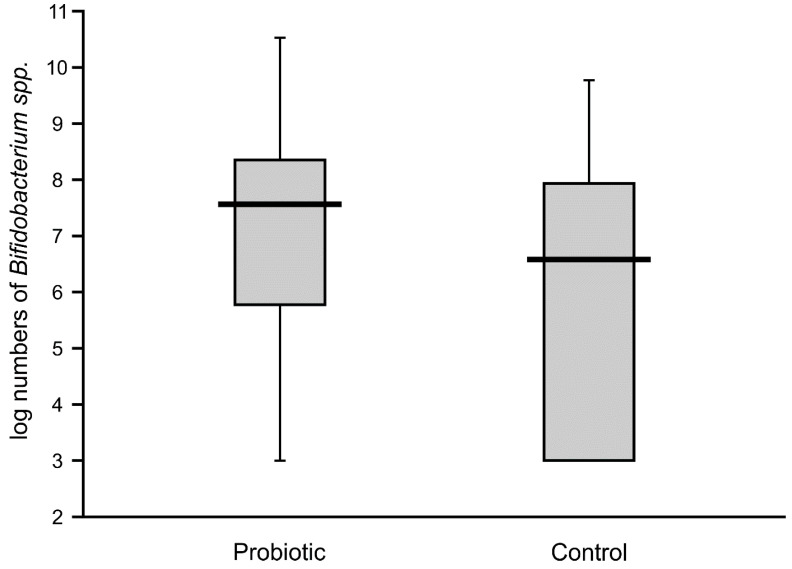
Abundance of bifidobacteria in feces obtained from the second sampling of neonates supplemented with probiotics vs. controls (Mann–Whitney test Z = 1.79087; *p* < 0.073315).

**Figure 6 nutrients-12-03128-f006:**
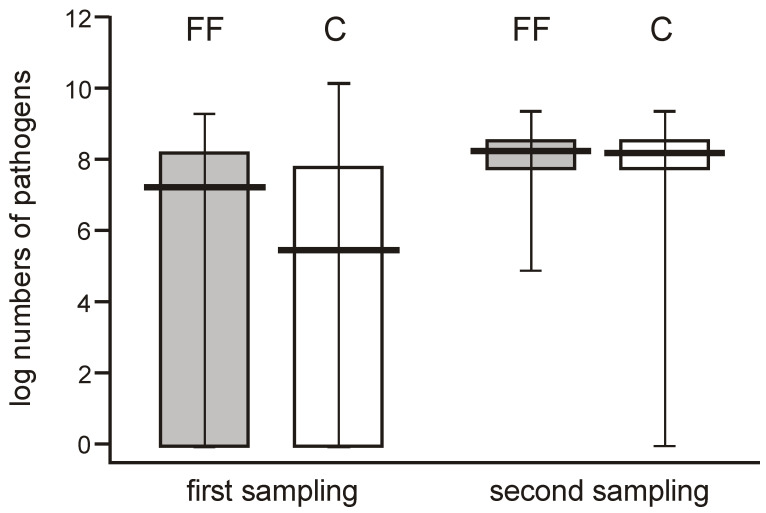
Changes in populations of the potentially pathogenic bacteria (*Clostridium*, *Staphylococcus*, *Enterococcus*, *Klebsiella*, and *Escherichia coli* taken together) found in 2 consecutive feces samplings obtained from neonates of the probiotic-supplemented and control group (Mann–Whitney test Z = 5.50712; *p* < 0.000001 and Z = −448466; *p* < 0.000007).

**Table 1 nutrients-12-03128-t001:** Characteristics of the study participants.

Total *n* = 148	Probiotic Group	Control Group
1st stool sample on 5th–6th day of life	*n* = 71	*n* = 77
Sex—males/females	*n* = 33/38	*n* = 37/40
Birth weight (g; mean values)	3560 ± 210	3480 ± 180
Breastfed (without any formula)	*n* = 43	*n* = 46
Antibiotics before or during labor (prophylaxis for surgical site infection (SSI) after cesarean section)	*n* = 71	*n* = 77
Excluded before second analysis (parents did not collect sample)	*n* = 13	*n* = 29
2nd stool sample	*n* = 58	*n* = 48
Sex—male	*n* = 27	*n* = 28

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
