# Peer review of "Effect of a Short-Time Probiotic Supplementation on the Abundance of the Main Constituents of the Gut Microbiota of Term Newborns Delivered by Cesarean Section—A Randomized, Prospective, Controlled Clinical Trial"

_nutrients, 2020, doi:10.3390/nu12103128_

Round 1

Reviewer 1 Report

Recent paper by Hurkala et al., focuses on the use of probiotics in the prevention of gut dysbiosis in newborns delivered by cesarean section. The objective of this study was to determine whether a probiotic supplement given to newborns delivered by CS alters the pattern of early gut colonization by bacteria.  In total, 150 newborns, born at 38–40 weeks gestational age and delivered by CS, were included in the study.

In the whole design of experiment, the read-out of the experiment is the content of certain bacteria species in the stool. To this read out additionally to probiotic treatment, there is also contribution of microbiota from breastfeeding and mums skin microbiota. How you explain this contribution?  There are huge variation in the detection of certain bacteria strain. It is weird that with 77 stool from newborns there is such a huge variations between samples. Possibly this variations can be impacted by difference in breast feeding, skin microbiota, different food uptake by breastfeeding mums and/or due to statistical approaches.

In the comparison 5-6 days between newborn treated with probiotic or not Lactobacillus spp. presence in probiotic group of newborns was higher. After month of the uptake of probiotic, the difference between control newborns and probiotic treated newborns difference in Lactobacillus spp. presence remains. Which is a good sign that the probiotic helped. The other good thing is that with or without a probiotic the level of pathogenic bacteria stay low. There is no significant differences detectable in pathogenic bacteria between the groups between control or probiotic treatment.

Bifidobacterium spp.were high in the stool of new borne that were up taking probiotic after 5-6 days. Although the levels of Bifidobacterium spp. after one months of uptake was similar to the control, which does not seem there is an impact of probiotic on this bactéria strain. The other good control would be stool from newborns delivered vaginally and potential this vaginally newborn treated with probiotic. This way the impact of probiotic would be compared and could be better evaluated.

Author Response

Recent paper by Hurkala et al., focuses on the use of probiotics in the prevention of gut dysbiosis in newborns delivered by cesarean section. The objective of this study was to determine whether a probiotic supplement given to newborns delivered by CS alters the pattern of early gut colonization by bacteria.  In total, 150 newborns, born at 38–40 weeks gestational age and delivered by CS, were included in the study.

In the whole design of experiment, the read-out of the experiment is the content of certain bacteria species in the stool. To this read out additionally to probiotic treatment, there is also contribution of microbiota from breastfeeding and mums skin microbiota. How you explain this contribution?  There are huge variation in the detection of certain bacteria strain. It is weird that with 77 stool from newborns there is such a huge variations between samples. Possibly this variations can be impacted by difference in breast feeding, skin microbiota, different food uptake by breastfeeding mums and/or due to statistical approaches.

Answer: It is known from several studies cited in our paper, that breastfeeding and skin are important sources of the microbiota acquisition by newborns very important in assembling neonatal gut microbiota. These sources are obviously highly variable from mother to mother and depending on many different factors. Thus, acquisition of the bacteria and gut subsequent colonization progresses differently in individual children which causes large variations in bacterial numbers present in feces observed also in our study.   Appropriate para was inserted to Discussion.    

In the comparison 5-6 days between newborn treated with probiotic or not Lactobacillus spp. presence in probiotic group of newborns was higher. After month of the uptake of probiotic, the difference between control newborns and probiotic treated newborns difference in Lactobacillus spp. presence remains. Which is a good sign that the probiotic helped. The other good thing is that with or without a probiotic the level of pathogenic bacteria stay low. There is no significant differences detectable in pathogenic bacteria between the groups between control or probiotic treatment.

Bifidobacterium spp. were high in the stool of new borne that were up taking probiotic after 5-6 days. Although the levels of Bifidobacterium spp. after one months of uptake was similar to the control, which does not seem there is an impact of probiotic on this bactéria strain. The other good control would be stool from newborns delivered vaginally and potential this vaginally newborn treated with probiotic. This way the impact of probiotic would be compared and could be better evaluated.

Answer: The fact that numbers of the bifidobacteria were similar in both groups in the one month after starting the probiotic supply, does not diminish the role of the early probiotic supplementation. As we have demonstrated in our previous study [13] based on the same laboratory methodology, a considerable proportion of the Lactobacillus and Bifidobacterium strains present in the neonates’ gut in this time after the probiotic application are identical with the applied strains. Concerning proposed studies on vaginally born children on probiotics, there are many reports on high numbers of the lactobacilli and bifidobacterial in their gut microbiota and therefore such a supplementation would be not accepted by our ethical committee.             

Reviewer 2 Report

Hurkala et al. present a prospective randomized control trial where they report the colonization of the neonatal stool with Lactobacillus rhamnosus KL53A and Bifidobacterium breve PB04 after a short term supplementation with these two strains.

General comments:

The authors state in the Introduction that the aim of probiotic supplementation after birth in general is to produce a microbiome more similar to a vaginally born child. However, they have not examined such microbiomes in the study. Therefore, the manuscript would benefit from a more elaborate description of the expexted microbiome as seen in vaginally bord children, as this will enable the readers to interpret the presented results more easily.

The manuscript would also benefit from a more precise rationale behind the research question. What aspects of the study are innovative, what new insights can the reader expect from this paper? In the current version, the distinction between the current study and the published literature is not clear.

Did the children benefit clinically from the supplementation (e.g. infection rate)?

Were all the cultured Lactobacilli and Bifidobacteria of the supplemented strain? And if yes, was this to be expected?

How many neonates were breast fed vs. formula fed? 

Did the authors collect samples before the children received their first dose of probiotics?

The figures in the supplementary file are in disarray. 

More specific comments: 

in line 74: better use written informed consent 

in line 91: Please provide a reference for the in-vitro tests performed with the probiotic strains

in line 108: the word "stomacher" is not familiar to me, is this a technical term or the name of the instrument?

in line 145: Please provide these characteristics in a table and add the percentage of breast and formula fed children. 

in line 153 and others: Please add a more commonly used numeric term to the used unit "logs per gram"

in line 158: is this 2 to 6 logs each or cummulative?

in line 172 and following: this sentence is hard to read and understand, please rephrase it.

in line 228 and 234: the authors describe other studies and use the phrases "their childred" and "their neonates". This is uncomfortable to read. The authors do not possess these children nor are they their parents. Please rephrase to something more neutral (eg. children in their study)

Author Response

Recent paper by Hurkala et al., focuses on the use of probiotics in the prevention of gut dysbiosis in newborns delivered by cesarean section. The objective of this study was to determine whether a probiotic supplement given to newborns delivered by CS alters the pattern of early gut colonization by bacteria.  In total, 150 newborns, born at 38–40 weeks gestational age and delivered by CS, were included in the study.

In the whole design of experiment, the read-out of the experiment is the content of certain bacteria species in the stool. To this read out additionally to probiotic treatment, there is also contribution of microbiota from breastfeeding and mums skin microbiota. How you explain this contribution?  There are huge variation in the detection of certain bacteria strain. It is weird that with 77 stool from newborns there is such a huge variations between samples. Possibly this variations can be impacted by difference in breast feeding, skin microbiota, different food uptake by breastfeeding mums and/or due to statistical approaches.

Answer: It is known from several studies cited in our paper, that breastfeeding and skin are important sources of the microbiota acquisition by newborns very important in assembling neonatal gut microbiota. These sources are obviously highly variable from mother to mother and depending on many different factors. Thus, acquisition of the bacteria and gut subsequent colonization progresses differently in individual children which causes large variations in bacterial numbers present in feces observed also in our study.   Appropriate para was inserted to Discussion.    

In the comparison 5-6 days between newborn treated with probiotic or not Lactobacillus spp. presence in probiotic group of newborns was higher. After month of the uptake of probiotic, the difference between control newborns and probiotic treated newborns difference in Lactobacillus spp. presence remains. Which is a good sign that the probiotic helped. The other good thing is that with or without a probiotic the level of pathogenic bacteria stay low. There is no significant differences detectable in pathogenic bacteria between the groups between control or probiotic treatment.

Bifidobacterium spp. were high in the stool of new borne that were up taking probiotic after 5-6 days. Although the levels of Bifidobacterium spp. after one months of uptake was similar to the control, which does not seem there is an impact of probiotic on this bactéria strain. The other good control would be stool from newborns delivered vaginally and potential this vaginally newborn treated with probiotic. This way the impact of probiotic would be compared and could be better evaluated.

The fact that numbers of the bifidobacteria were similar in both groups in the one month after starting the probiotic supply, does not diminish the role of the early probiotic supplementation. As we have demonstrated in our previous study [13] based on the same laboratory methodology, a considerable proportion of the Lactobacillus and Bifidobacterium strains present in the neonates’ gut in this time after the probiotic application are identical with the applied strains. Concerning proposed studies on vaginally born children on probiotics, there are many reports on high numbers of the lactobacilli and bifidobacterial in their gut microbiota and therefore such a supplementation would be not accepted by our ethical committee.             

Round 2

Reviewer 2 Report

The authors improved the manuscript. However, some additional comments arose with the revision:

  1. In table 1, information about sex, birthweigth, formula/brest feeding, maternal antibiotic treatment before labor, time of first sampling, and other relevant patients characteristics are missing. Instead Table 1 reproduces the flow chart in figure. please revise.
  2. The authors state in their reply letter that the characterization of the microbiome before the probiotic intervention would have been "useless". I disagree. If these measurements were not compatible with their study protocol, this needs to be discussed as a limitation of the study. 
  3. Limitations in general should be added to the discussion.  
  4. The conclusions are still not entirely supported by the results. Since the study does not include vaginally born children, the comparison of the probiotic fed children is not possible. Also, the study has not established the optimal time point for supplementation. Please revise.

Author Response

Comments and Suggestions for Authors

The authors improved the manuscript. However, some additional comments arose with the revision:

  1. In table 1, information about sex, birthweigth, formula/brest feeding, maternal antibiotic treatment before labor, time of first sampling, and other relevant patients characteristics are missing. Instead Table 1 reproduces the flow chart in figure. Please revise.

Answer: Table 1 was changed, and the data indicated by the reviewer 2 were added. Still, since we found no statistical differences between the two groups, as written in the text, we think that insertion of these data into the separate table brings no added value.   

  1. The authors state in their reply letter that the characterization of the microbiome before the probiotic intervention would have been "useless". I disagree. If these measurements were not compatible with their study protocol, this needs to be discussed as a limitation of the study.

Answer: As stressed in our first reply, it was neither our intention, while planning the study, nor when constructing our study protocol, to characterize gut microbiota of the neonates born by CS but to monitor populations of the lactic acid bacteria vs. potential pathogens in the neonates after probiotic intervention to mimic natural colonization with these bacteria in vaginally delivered children. To our opinion, including vaginally delivered neonates to our study would be just repeating dozens in already published studies on natural colonization. Moreover, such approach could be a source of errors from epidemiological point of view, since mothers and their CS-delivered children stay for several days in a different clinic then those born vaginally. A paragraph on limitations of the study was added to Discussion.   

  1. Limitations in general should be added to the discussion.

Answer: A paragraph on limitations of the study was added to Discussion

  1. The conclusions are still not entirely supported by the results. Since the study does not include vaginally born children, the comparison of the probiotic fed children is not possible. Also, the study has not established the optimal time point for supplementation. Please revise.

Answer: Again, there was no reason to study neonates already colonized with vaginally derived lactic acid bacteria. Moreover, I’m afraid that such a control would be not accepted by our ethical committee. Optimal time for supplementation was derived from our previous studies as written in the text. Also, since vaginally born babies are colonized during the labor, optimal time for artificial colonization should be as close as possible to natural conditions. Conclusion was modified.
